# A hidden vulnerable population: Young children up-to-date on vaccine series recommendations except influenza vaccines

**William K. Bleser**[1¤]\*, **Daniel A. Salmon**[2], **Patricia Y. Miranda**[3]

**1** Robert J. Margolis, MD, Center for Health Policy, Duke University, Washington, DC, United States of America, **2** Department of International Health, Johns Hopkins Bloomberg School of Public Health, Baltimore, MD, United States of America, **3** Department of Health Policy and Administration, Pennsylvania State University, University Park, PA, United States of America

¤ Current address: Work was completed while at Department of Health Policy and Administration, Pennsylvania State University, University Park, PA, United States of America

\* william.bleser@duke.edu

**Data Availability Statement:** The data are owned by the National Center for Health Statistics (NCHS), of the Centers for Disease Control and Prevention (CDC), of the United States Department of Health

## Abstract

Very young children (under 2 years old) have high risk for influenza-related complications. Children 6 months or older in the US are recommended to receive influenza vaccination annually, yet uptake is substantially lower than other routinely-recommended vaccines. Existing nationally-representative studies on very young child influenza vaccine uptake has several limitations: few examine provider-verified influenza vaccination (relying on parental report), few contain parental vaccine attitudes variables (known to be crucial to vaccine uptake), and none to our knowledge consider intersectionality of social disadvantage nor how influenza vaccine determinants differ from those of other recommended vaccines. This nationally-representative study examines provider-verified data on 7,246 children aged 6–23 months from the most recent (2011) National Immunization Survey to include the restricted Parental Concerns module, focusing on children up-to-date on a series of vaccines (the 4:3:1:3:3:1:4 series) but not influenza vaccines ("hidden vulnerability to influenza"). About 71% of children were up-to-date on the series yet only 33% on influenza vaccine recommendations by their second birthday; 44% had hidden vulnerability to influenza. Independent of parental history of vaccine refusal and a myriad of health services use factors, no parental history of delaying vaccination was associated with 7.5% (2.6–12.5) higher probability of hidden vulnerability to influenza despite being associated with 15.5% (10.8–20.2) lower probability of being up-to-date on neither the series nor influenza vaccines. Thus, parental compliance with broad child vaccine recommendations and lack of vaccine hesitancy may not indicate choice to vaccinate children against influenza. Examination of intersectionality suggests that maternal college education may not confer improved vaccination among non-Hispanic Black and Hispanic children despite that it does for non-Hispanic White children. Policymakers and researchers from public health, sociology, and other sectors need to collaborate to further examine how vaccine hesitancy and intersectional social disadvantage interact to affect influenza vaccine uptake in young US children.

and Human Services (HHS). Others can obtain access to these data at a Federal Statistical Research Data Center in the same manner that the authors did. To do so, they must submit a proposal to the NCHS for the restricted data, have the proposal approved by the NCHS Research Ethics Review Board, and then successfully follow guidance for receiving Special Sworn Status (SSS) from the United States Census Bureau. Then they will receive access to the data at a Federal Statistical Research Data Center. Interested researchers can email the NCHS Research Data Center at rdca@cdc.gov to ask question about the data or the process for obtaining the data. More information on the application process for obtaining the data can also be found at www.cdc.gov/rdc/index.htm.

**Funding:** There was no external funding received for this study. We acknowledge indirect support provided by Pennsylvania State University's Department of Health Policy and Administration, Demography program, and Population Research Institute. The Population Research Institute is supported by an infrastructure grant from NIH (2R24HD041025-11). This publication was also supported, in part, by Grant UL1 TR000127 and KL2 TR000126 from the National Center for Advancing Translational Sciences (NCATS). The funders provided support in the form of salaries or infrastructure for authors WKB and PYM, but did not have any additional role in the study design, data collection and analysis, decision to publish, or preparation of the manuscript. The specific roles of these authors are articulated in the 'author contributions' section.

**Competing interests:** William K. Bleser discloses past consulting fees from Merck unrelated to this research. Daniel A. Salmon discloses consulting fees and research grants from Merck, Pfizer, and Walgreens unrelated to this research. No other financial disclosures were reported. This does not alter our adherence to PLOS ONE policies on sharing data and materials.

# Introduction

Children under the age of 5 years ("young") and especially under 2 years ("very young") are high risk for influenza complications simply because of their age, even if otherwise healthy. [1,2] They have increased risk of influenza-related hospitalizations, and doctor, urgent care, and emergency department visits, [3,4] comprising a substantial portion of total US influenza morbidity. [5] Influenza in children also affects family members and caregivers, [6] causing substantial parental work absenteeism, [7] and community epidemics. [8]

Influenza vaccination is the most effective preventive measure [9] and the US Centers for Disease Control and Prevention (CDC) routinely recommends it for all persons 6 months and older. [10] Influenza vaccines continually demonstrate a great safety profile, [11] and though their effectiveness varies annually, in children they prevent doctor visits, [12] febrile illnesses, [13] hospitalizations, [14] and randomized trials show high pooled efficacy of the live, attenuated vaccine (83% relative reduction of influenza risk) for children <8 years old. [15] Moreover, there is building evidence that vaccinating children against influenza has benefits extending to other adults in the household (for example, by preventing work loss [16–20]). Influenza vaccines have been increasingly affordable and available to children through public programs [21] and because the Affordable Care Act requires new health plans to cover all routinely-recommended preventive services without cost-sharing. [22]

Influenza vaccination uptake in young US children, however, remains sub-optimal. National annual uptake recently peaked at 73% during the 2018/2019 influenza season but has generally plateaued around 70% over the last decade of influenza seasons–as low as 43% in some states–representing millions of unvaccinated children. [23] Further, "complete uptake" as defined by the CDC–receiving the appropriate number of influenza vaccinations for the child's age and birthdate–is generally much lower in young children. [24]

By contrast, complete uptake of other routinely-recommended vaccines is much higher. In the most recent published estimates (2017), the percent of children 19–35 months old up-to-date (UTD) on other recommendations was: 83.2% for 4+ diphtheria-tetanus-acellular pertussis vaccine doses, 92.7% for 3+ poliovirus vaccine doses, 91.5% for 1+ measles-mumps-rubella vaccine doses, 80.7% for 3+ *Haemophilus influenza* type B vaccine doses, 91.4% for 3+ Hepatitis B vaccine doses, 91.0% for 1+ varicella vaccine doses, and 82.4% for 4+ pneumococcal conjugate vaccine doses. [25] Moreover, the percentage UTD on *all* of these other recommendations (the "4:3:1:3:3:1:4" series) is 70.4%. [25]

Research on determinants of uptake for influenza vaccination in the US, however, is limited, tending to focus on adult (particularly elderly) populations, and substantially less on children. [26,27] Though an unpublished literature review [28] and published studies of other vaccines [29,30] provide theoretical and empirical foundations of determinants to consider, there are three limitations.

First, existing studies and frameworks have limited generalizable to the general pediatric population.

Second, there is no comparison of determinants of being UTD on influenza vaccines vs. other vaccines. This is an important research gap; that the 19-shot, 7-vaccine series (4:3:1:3:3:1:4) uptake rate is comparable to the recent 2018/2019 single-season peak in influenza vaccine uptake in young children indicates unique mechanisms affect parents' decisions to vaccinate their child against influenza relative to every other routinely-recommended childhood vaccine.

Third, to our knowledge, no nationally-representative studies utilized a conceptual framework to ground their selection of covariates. As a result, the literature does not systematically consider and adjust for many important constructs, notably vaccine-related parental

perceptions. Moreover, no studies consider interacting effects of disadvantaged social statuses, an important limitation potentially obscuring health differences and impairing efforts to reduce health disparities. [31] Intersectionality theory posits that social statuses like race/ethnicity, gender, and social class cannot be disaggregated as they reinforce each other in producing and maintaining health outcomes across the life span. [32–35]

This study has the goal of replicating prior studies examining determinations of influenza vaccine uptake of very young children while directly addressing the three aforementioned sets of limitations. To do so, this study uses a nationally-representative sample of very young children in the US that includes provider-verified vaccination status and constructs across all domains noted in the literature, including federally-restricted variables about parental attitudes of vaccination and accounting for intersectionality. It examines determinants of a newly-identified vulnerable population: those with "hidden vulnerability to influenza"–i.e., children UTD on a wide variety of vaccine recommendations (the 4:3:1:3:3:1:4 series) except influenza.

## Methods

### Data source

Data come from the 2011 National Immunization Survey (NIS), which includes the most recent Parental Concerns (PC) module, a restricted supplement containing important vaccine-related parental perception variables [36]. The NIS is a serial, cross-sectional survey that has monitored child vaccination uptake since 1994. [37] The target population is children 19–35 months in US households. [38] The PC module variables were merged with publicly-accessible NIS variables by National Center for Health Statistics (NCHS) analysts and accessed by the authors at the Penn State Federal Statistical Research Data Center, a Census Bureau facility housed at the Pennsylvania State University meeting all physical and information security requirements for federally-restricted data.

The research protocol was reviewed by both the NCHS Research Ethics Review Board and the Pennsylvania State University Institutional Review Board and deemed not human research.

The NIS uses random digit dialing methodology to identify households containing target children and interviews a knowledgeable adult. With consent, the NIS contacts the child's health care provider(s) by mail to request vaccination information from the child's medical records; 79.5% and 75.0% of landline and cell phone cases gave consent; 95.2% and 93.8% of their providers returned the questionnaires. The 2011 public-use file contains 26,741 children with completed interviews, and 19,144 with provider-verified data (excluding the Virgin Islands). Overall, the CASRO response rate was 61.6% (72.3% of which had adequate provider data). [38] Of the 19,144 children with adequate provider-verified data, 13,358 (69.8%) received the restricted PC module, and 12,559 (94.0%) completed it (unpublished NCHS data that the authors obtained via correspondence with NCHS analysts).

### Dependent variable

Two binary NIS variables were used to construct the three dependent variables used in this study. The first is complete influenza vaccination–that is, whether the child received the full number of seasonal influenza vaccines given the number of influenza seasons they have experienced by their second birthday and when the survey was administered (children not 6–23 months of age during the span of September 1 to December 31 are "not eligible;" see Section 7.8.1 and Table 7 of the survey user's guide [38]). The second variable captures whether the child is UTD on the 4:3:1:3:3:1:4 series. The three binary dependent variables used in this

study are combinations of these two NIS variables–being UTD on: (1) "both" requirements; (2) "series but not influenza" requirements; and (3) "neither" requirement. These terms are used throughout the paper. The focus of this study is on the "series but not influenza" outcome in order to address the gap of identifying determinants that uniquely predict children UTD on a wide variety of vaccine recommendations except influenza in order to predict "hidden" vulnerability to influenza.

## Determinants of influenza vaccination

Vaccination is the use of a health service, so selection of determinants can be grounded in Andersen's model of health services use, [39] which divides determinants into three factors: (1) *predisposing* (e.g., child's race/ethnicity, parental vaccine attitudes and beliefs); (2) *enabling* (e.g., family income, health insurance); and (3) *need* (e.g., functional state, need for medical care). The model also accounts intermediate-level health behaviors influencing health services use (e.g., personal health practices). Andersen's model has been used across multiple health-care system sectors in the context of a variety of diseases. [40] All NIS variables pertinent to this model or prior vaccine literature were included as described below (see Table 1 for more detail):

Seven variables represent contextual-level factors (family- or medical practice-level) predisposing, enabling, or creating need for influenza vaccination and other health services use:

1. *mother's education* [41–43];

2. *mother's age* [44];

3. *mother's marital status*;

4. *household language* [44];

5. *housing arrangement*;

6. *area of residence*; and

7. *provider facility type* [43,44]

Seven variables represent parental perceptions and beliefs surrounding vaccines and vaccine-preventable diseases. The Parental Concerns module data are restricted and not contained in the public use dataset, but these variables were obtained by the authors and analyzed in a Research Data Center for his study. However, the survey instrument is publicly available online [36]. Questions 1–5 below ask parents to rate the statement on a scale of 0–10 where 0 is "strongly disagree" and 10 is "strongly agree." Questions 6 and 7 below ask parents if they have ever refused or delayed getting their child vaccinated (binary question):

1. *vaccines are necessary to protect child health* [26,45–50];

2. *vaccines do a good job at preventing their diseases* [26,45–50];

3. *vaccines are safe* [45–47,51];

4. *vaccine-preventable diseases are serious and can hurt children* [26,46,52];

5. *strength of physician vaccine recommendation* [27,41,45–50,52–56];

6. *history of refusing their child's vaccines*; and

7. *history of delaying their child's vaccines*.

Five variables represent individual (child)-level factors:

**Table 1. Descriptive statistics of study population, U.S children aged 6–23 months old (N = 7,246), 2011 NIS.**

| Variable | Percent | N |
|---|---|---|
| **Outcome variables** | | |
| Total up-to-date on influenza vaccine(s) at 24 months old | | |
| No | 66.7 | 4602 |
| Yes | 33.3 | 2644 |
| Total up-to-date on 4:3:1:3:3:1:4 vaccine series | | |
| No | 28.9 | 2048 |
| Yes | 71.1 | 5198 |
| *Up-to-date on BOTH influenza vaccine(s) AND 4:3:1:3:3:1:4 vaccine series | | |
| No | 72.5 | 5042 |
| Yes | 27.6 | 2204 |
| **Up-to-date on ONLY 4:3:1:3:3:1:4 vaccine series; not influenza vaccine(s) | | |
| No | 56.5 | 4252 |
| Yes | 43.5 | 2994 |
| Up-to-date on ONLY influenza vaccine(s); not 4:3:1:3:3:1:4 vaccine series | | |
| No | 94.3 | 6806 |
| Yes | 5.8 | 440 |
| *Up-to-date on NEITHER influenza vaccine(s) NOR 4:3:1:3:3:1:4 vaccine series | | |
| No | 76.8 | 5638 |
| Yes | 23.2 | 1608 |
| **Independent variables** | **Percent** | **N** |
| Child's sex | | |
| Female | 48.2 | 3503 |
| Male | 51.8 | 3743 |
| Child's race/ethnicity | | |
| Non-Hispanic White only | 50.8 | 4629 |
| Non-Hispanic Black only | 12.5 | 690 |
| Non-Hispanic other or multiple race | 9.2 | 757 |
| Hispanic | 27.5 | 1170 |
| Child's first-born status | | |
| First born | 40.6 | 2399 |
| Not first born | 59.4 | 4847 |
| Child ever received benefits from the *Women, Infants, and Children* program | | |
| No | 47.8 | 4325 |
| Yes | 52.2 | 2921 |
| Child uninsured | | |
| No | 91.9 | 6755 |
| Yes | 8.1 | 491 |
| Mother's education | | |
| Less than a college graduate | 63.8 | 3706 |
| College graduate | 36.2 | 3540 |
| Mother's age group | | |
| ≤19 years | 2.6 | 121 |
| 20–29 years | 41.6 | 2144 |
| ≥30 years | 55.9 | 4981 |
| Mother's marital status | | |
| Married | 67.9 | 5506 |
| Never married, widowed, divorced, separated, or deceased | 32.1 | 1740 |
| Language | | |

(*Continued*)

**Table 1.** (Continued)

| Variable | Percent | N |
|---|---|---|
| **Outcome variables** | | |
| English | 87.0 | 6712 |
| Spanish or other | 13.0 | 534 |
| Housing arrangement | | |
| Owned or being bought | 57.1 | 5153 |
| Rented | 39.5 | 1889 |
| Other arrangement | 3.4 | 204 |
| Provider facility type | | |
| Public/WIC | 11.4 | 751 |
| Hospital | 10.5 | 836 |
| Private | 60.5 | 4464 |
| Military/other facilities | 3.9 | 242 |
| Mixed | 13.7 | 953 |
| Child was ever breastfed or fed breast milk | | |
| No | 22.4 | 1406 |
| Yes | 77.6 | 5840 |
| Parent ever refused or decided not to have their child vaccinated | | |
| No | 84.6 | 6052 |
| Yes | 15.4 | 1194 |
| Parent ever delayed or put off having their child vaccinated | | |
| No | 66.6 | 4852 |
| Yes | 33.4 | 2394 |
| | **Mean (SD)** | **N** |
| Parent belief that vaccines are necessary to protect children's health | 9.4 (1.3) | 7246 |
| Parent belief that vaccines do a good job at preventing their diseases | 9.1 (1.6) | 7246 |
| Parent belief that vaccines are safe | 8.3 (2.1) | 7246 |
| Parent belief that vaccine-preventable diseases are serious and can hurt children | 9.2 (2.1) | 7246 |
| Parent perception of strength of physician's vaccine recommendation | 9.3 (1.7) | 7246 |

Source: 2011 National Immunization Survey (NIS) data, children represented in the Parental Concerns module with provider-verified vaccination data and eligible for the influenza vaccination up-to-date question who are not missing any covariates. Means and percentages weighted to be nationally-representative. N un-weighted to show actual number of observations in each cell. For the last 5 covariates (parent beliefs/perceptions), the scale is 0–10 where 0 is disagree and 10 is agree.

*Comparator outcome variables examined in this study

**Main outcome of interest in this study, "series but not influenza" (i.e., "hidden vulnerability to influenza")

1. *sex*;

2. *race/ethnicity* [24,43,44]);

3. *first born status*;

4. *current receipt of Women, Infants, and Children (WIC) benefits*; and

5. whether the child was *uninsured* at any time during the year [52].

   One variable represents the child's personal health practices–*whether they were ever breast fed/fed breast milk*. A variable for family income was considered but exhibited concerns of multicollinearity and thus was excluded.

## Study population

Respondents were eligible for the study if they: (1) had provider-verified data (NIS-defined eligibility for the outcome variables; also addresses recall bias gap in other literature); (2) were not ineligible for the influenza UTD variable by age at survey date (NIS-defined eligibility for the outcome variables); and (3) received the Parental Concerns module (8,065 total eligible children). Complete case analysis was performed; 89.8% of the eligible sample were complete cases across all variables (N = 7,246). Complete case status was neither associated with the main outcome ("series but not influenza" UTD status), nor 15 of 20 covariates. Because complete case status was only slightly associated with 5 of the 20 covariates, missingness was not completely at random (a key assumption for ruling out multiple imputation for dealing with missingness). Moreover, the large size of the complete case sample, relatively low complete case missingness, and lack of association between complete case status and outcome of interest all suggest complete case analysis to be less biased than other methods of dealing with missingness such as multiple imputation, [57] so complete case analysis was performed.

## Analysis

We performed three sets of analyses. First, we examined variation in each vaccine UTD outcome by independent variables of interest and covariates. Second, we performed regression analyses to examine the relationship between vaccine UTD outcomes and key independent variables controlling for covariates and using interaction terms to examine intersectionality. Third, we examined model-predicted outcome probabilities and graphed their patterns to interpret the intersectional results. Those three sets of analyses are described in detail below:

First, bivariate associations between the three UTD outcomes and all determinants (variables) were examined.

Second, each outcome was then regressed onto all determinants, including interaction terms for all combinations of child's race/ethnicity, mother's education, and mother's marital status to incorporate intersectionality. Logistic regression is often used to examine bivariate outcomes, though we use Linear Probability Model (LPM) regression–Ordinary Least Squares regression of a binary outcome–because logistic regression does not produce straightforward interpretation of interaction terms. [58,59] Further, LPM regression is motivated by the literature [60–62] and its coefficients are easily interpreted as changes in the probability of observing the "1" binary response associated with unit changes in explanatory variables.

Third, given interaction term coefficients are not directly interpretable, [63] model-predicted marginal probabilities of UTD status among all interaction term subgroups were calculated and graphed. Analyzing double and triple interaction terms can be complicated to interpret from just the numbers, so we graphed the predicted probability to visually compare changes in the outcome of interest among all interaction term subgroups in a side-by-side manner.

All analyses were performed using Stata/SE 13.1 statistical software [64] and use Stata's *svy* commands to apply NIS-provided sample weights to generate national-representative estimates adjusted for complex survey design, ratio, non-response, post-stratification adjustments, and heteroscedasticity.

## Results

Table 1 contains weighted descriptive statistics of the complete case sample. By their second birthday, 33% of children were UTD on influenza vaccinations, and 71% were UTD on the 4:3:1:3:3:1:4 series. The cross-section of these variables (this study's outcomes) reveals that 27%

were UTD on both, 23% were UTD on neither, and 44% were UTD on the series but not influenza vaccines (again, the latter variable being the main interest of this study).

Table 2 provides weighted bivariate correlations (i.e., not adjusted for any other variables) between the three UTD outcomes and each covariate. There were several determinants associated with vulnerability across all of the UTD outcomes (see the shaded gray cells), but two findings were unique to "series but not influenza"–children in households speaking Spanish or another language (9 percentage points more likely than English households to have hidden vulnerability to influenza, p = 0.023), and never delaying vaccination (8 percentage points more likely than ever delaying to have hidden vulnerability to influenza, p = 0.003).

Table 3 shows weighted results from LPM regression of the "series but not influenza" outcome onto all determinants (i.e., adjusted for all variables), including interaction terms. Comparing all columns, several patterns emerge (see the shaded gray cells). Ever refusing vaccination was associated with 9.9 percentage points (95% confidence interval (CI): 4.2–15.7) *higher* probability of "series but not influenza" (hidden vulnerability to influenza) despite that ever delaying (not necessarily refusing) was associated with 7.5 percentage points (95% CI 2.6–12.5) *lower* probability of "series but not influenza." The direction of the delay finding was unexpected from what was observed in the other two outcomes (S1 Table).

Some interaction term coefficients in Table 3 related to combinations of mother's education and child's race/ethnicity were significant and the direction of the "series but not influenza" coefficients were also different than what would be expected from the other two outcomes (S1 Table). These warrant exploration of patterns among the interaction term variables and suggest that intersectionality matters for hidden vulnerability to influenza. Accordingly, to interpret interaction term coefficients, Table 4 shows weighted, predicted probabilities of each UTD outcome among all possible combinations of interaction terms. There were no significant interaction term coefficients involving mother's marital status in the "series but not influenza" outcome from Table 3 and no significant differences in predicted probabilities of intersectional subgroups in Table 4. There were also no significant differences within predicted probabilities of each lone intersectional construct (see Fig 1).

However, examination of the predicted probabilities of "series but not influenza" among child's race/ethnicity*mother's education subgroups elucidates why there were significant interactions terms observed in Table 3. First, Hispanic children with college-educated mothers have higher probability (0.565: 95% CI 0.447–0.683) of "series but not influenza" than non-Hispanic White children with college-educated mothers (0.344: 0.291–0.396) despite that the former had one of the lowest predicted probabilities of the "both" outcome (S2 Table); this indicates that a unique identifier of hidden vulnerability for influenza is in Hispanic children with college-educated mothers. Second, examining the graphical representation of this relationship (Fig 2) shows that mother's education is associated with reduced "series but not influenza" probability among non-Hispanic White and non-Hispanic Other children but increased probability for non-Hispanic Black and Hispanic children.

Finally, the triple-interaction term coefficients were examined to further explore the above intersectionality finding. In Table 4, Hispanic children with married, college-educated mothers were significantly more likely to be in the "series but not influenza" group (0.603: 0.489–0.717) than non-Hispanic White children with college-educated mothers regardless of whether the mother was married (0.366: 0.319–0.413) or not (0.295: 0.166–0.424). Visualizing this in Fig 3, which stratifies Fig 2 by mother's marital status, a clear trend emerges: the patterns seen among married mothers (top panel of Fig 3) closely mimic the unstratified relationship depicted in Fig 2. Looking at the pattern among mothers never married, widowed, divorced, separated, or deceased (bottom panel of Fig 3), however, reveals a divergence in Hispanic women: attainment of a college degree is associated with hidden vulnerability to influenza

**Table 2. Correlates of vaccination up-to-date variables, U.S children aged 6–23 months old (N = 7,246), 2011 NIS.**

| | Up-to-date status (combinations of seasonal influenza and the 4:3:1:3:3:1:4 series) | | | | | | | | |
| --- | --- | --- | --- | --- | --- | --- | --- | --- | --- |
| | "BOTH" Both flu and 4:3:1:3:3:1:4 series 72.5% 27.6% | | | "SERIES BUT NOT FLU" 4:3:1:3:3:1:4 series, not flu 56.5% 43.5% | | | "NEITHER" Neither flu, 4:3:1:3:3:1:4 series 76.8% 23.2% | | |
| | No % | Yes % | p | No % | Yes % | p | No% | Yes % | p |
| **Child's sex** | | | | | | | | | |
| Female | 72.7 | 27.3 | 0.8430 | 57.4 | 42.6 | 0.4609 | 75.8 | 24.2 | 0.3583 |
| Male | 72.3 | 27.8 | | 55.6 | 44.4 | | 77.8 | 22.2 | |
| **Child's race/ethnicity** | | | | | | | | | |
| Non-Hispanic White only | 68.4 | 31.6 | 0.0002 | 59.3 | 40.7 | 0.0220 | 78.6 | 21.4 | 0.0113 |
| Non-Hispanic Black only | 82.5 | 17.5 | | 56.9 | 43.1 | | 68.0 | 32.0 | |
| Non-Hispanic other/multiple race | 68.9 | 31.1 | | 57.9 | 42.2 | | 78.8 | 21.3 | |
| Hispanic | 76.6 | 23.4 | | 50.5 | 49.5 | | 77.0 | 23.0 | |
| **Child's first-born status** | | | | | | | | | |
| First born | 70.1 | 29.9 | 0.0572 | 54.1 | 45.9 | 0.1026 | 82.1 | 18.0 | 0.0002 |
| Not first born | 74.1 | 25.9 | | 58.1 | 41.9 | | 73.3 | 26.7 | |
| **Child ever received WIC benefits** | | | | | | | | | |
| No | 65.9 | 34.1 | <0.0001 | 59.3 | 40.7 | 0.0210 | 81.3 | 18.7 | 0.0001 |
| Yes | 78.4 | 21.6 | | 53.9 | 46.1 | | 72.8 | 27.2 | |
| **Child uninsured** | | | | | | | | | |
| No | 71.8 | 28.2 | 0.0270 | 56.4 | 43.6 | 0.9200 | 77.7 | 22.3 | 0.0099 |
| Yes | 79.6 | 20.4 | | 56.9 | 43.1 | | 67.4 | 32.6 | |
| **Mother's education** | | | | | | | | | |
| Less than a college graduate | 77.3 | 22.7 | <0.0001 | 54.4 | 45.6 | 0.0141 | 73.5 | 26.5 | <0.0001 |
| College graduate | 63.9 | 36.1 | | 60.0 | 40.0 | | 82.8 | 17.2 | |
| **Mother's age group** | | | | | | | | | |
| ≤19 years | 83.2 | 16.8 | <0.0001 | 44.0 | 56.0 | 0.1680 | 73.3 | 26.7 | 0.0010 |
| 20–29 years | 77.8 | 22.2 | | 55.6 | 44.4 | | 72.5 | 27.5 | |
| ≥30 years | 68.0 | 32.0 | | 57.7 | 42.3 | | 80.3 | 19.8 | |
| **Mother's marital status** | | | | | | | | | |
| Married | 69.0 | 31.0 | <0.0001 | 58.0 | 42.0 | 0.0579 | 79.1 | 20.9 | 0.0019 |
| Never married, widowed, divorced, separated, or deceased | 79.8 | 20.2 | | 53.2 | 46.8 | | 72.1 | 27.9 | |
| **Language** | | | | | | | | | |
| English | 72.1 | 27.9 | 0.445 | 57.6 | 42.4 | 0.0228 | 76.3 | 23.7 | 0.2179 |
| Spanish or other | 75.0 | 25.0 | | 48.6 | 51.4 | | 80.3 | 19.7 | |
| **Housing arrangement** | | | | | | | | | |
| Owned or being bought | 69.5 | 30.5 | 0.0017 | 56.6 | 43.4 | 0.7133 | 80.0 | 20.0 | 0.0013 |
| Rented | 75.6 | 24.4 | | 56.6 | 43.4 | | 73.0 | 27.0 | |
| Other arrangement | 85.3 | 14.7 | | 41.4 | 48.6 | | 68.6 | 31.4 | |
| **Provider facility type** | | | | | | | | | |
| Public/WIC | 82.0 | 18.0 | 0.0003 | 53.2 | 46.8 | 0.5188 | 67.1 | 32.9 | <0.0001 |
| Hospital | 76.6 | 23.4 | | 58.0 | 42.0 | | 75.9 | 24.1 | |
| Private | 69.6 | 30.4 | | 57.6 | 42.4 | | 78.9 | 21.1 | |
| Military/other facilities | 85.5 | 14.6 | | 58.7 | 41.3 | | 57.6 | 42.4 | |
| Mixed | 70.2 | 29.8 | | 52.3 | 47.7 | | 82.2 | 17.8 | |
| **Child was ever breastfed or fed breast milk** | | | | | | | | | |
| No | 79.5 | 20.5 | <0.0001 | 53.3 | 46.7 | 0.1333 | 73.3 | 26.7 | 0.0500 |
| Yes | 70.4 | 29.6 | | 57.4 | 42.6 | | 77.9 | 22.1 | |

(*Continued*)

**Table 2.** (Continued)

| | Up-to-date status (combinations of seasonal influenza and the 4:3:1:3:3:1:4 series) | | | | | | | | |
|---|---|---|---|---|---|---|---|---|---|
| | "BOTH" Both flu and 4:3:1:3:3:1:4 series 72.5% 27.6% | | | "SERIES BUT NOT FLU" 4:3:1:3:3:1:4 series, not flu 56.5% 43.5% | | | "NEITHER" Neither flu, 4:3:1:3:3:1:4 series 76.8% 23.2% | | |
| | No % | Yes % | p | No % | Yes % | p | No% | Yes % | p |
| Parent ever refused/decided not to have their child vaccinated | | | | | | | | | |
| No | 70.1 | 29.9 | <0.0001 | 57.4 | 42.6 | 0.0340 | 78.5 | 21.5 | <0.0001 |
| Yes | 85.4 | 14.6 | | 51.1 | 48.9 | | 67.8 | 32.3 | |
| Parent ever delayed or put off having their child vaccinated | | | | | | | | | |
| No | 69.3 | 30.7 | 0.0003 | 53.9 | 46.1 | 0.0032 | 82.6 | 17.4 | <0.0001 |
| Yes | 78.7 | 21.3 | | 61.6 | 38.4 | | 65.4 | 34.6 | |
| | No mean (se) | Yes Mean (se) | p | No mean (se) | Yes Mean (se) | p | No mean (se) | Yes Mean (se) | p |
| Parent believes vaccines are necessary to protect children's health | 9.33 (0.03) | 9.58 (0.04) | <0.0001 | 9.36 (0.04) | 9.46 (0.04) | 0.0775 | 9.50 (0.03) | 9.08 (0.08) | <0.0001 |
| Parent believes vaccines do a good job at preventing their diseases | 9.02 (0.05) | 9.21 (0.07) | 0.0252 | 9.01 (0.06) | 9.15 (0.05) | 0.0658 | 9.18 (0.04) | 8.73 (0.11) | 0.0001 |
| Parent believes vaccines are safe | 8.16 (0.06) | 8.63 (0.07) | <0.0001 | 8.26 (0.06) | 8.34 (0.09) | 0.4618 | 8.43 (0.06) | 7.83 (0.11) | <0.0001 |
| Parent believes vaccine-preventable diseases are serious and can hurt children | 9.13 (0.07) | 9.27 (0.09) | 0.2228 | 9.20 (0.07) | 9.12 (0.08) | 0.4880 | 9.20 (0.06) | 9.07 (0.14) | 0.4125 |
| Parent perceived strength of physician vaccine recommendation | 9.32 (0.04) | 9.41 (0.12) | 0.5055 | 9.33 (0.07) | 9.36 (0.06) | 0.7548 | 9.37 (0.06) | 9.27 (0.07) | 0.2852 |

Source: 2011 National Immunization Survey (NIS) data, children represented in the Parental Concerns module with provider-verified vaccination data and eligible for the influenza vaccination up-to-date question who are not missing any covariates. Means and percentages weighted to be nationally-representative. For the last 5 covariates (parent beliefs/perceptions), the scale is 0–10 where 0 is disagree and 10 is agree. Shaded cells indicate most vulnerable groups among those with statistically significant differences in each UTD outcome.

**Table 3. Change in predicted probabilities of up-to-date vaccine status, multivariate linear probability model regression, U.S children aged 6–23 months old (N = 7,246), 2011 NIS.**

| | Up-to-date status: "SERIES BUT NOT FLU" 4:3:1:3:3:1:4 series, not flu | |
|---|---|---|
| | ΔPr. | 95% CI |
| **Child's race/ethnicity** (ref: non-Hispanic White) | | |
| Non-Hispanic Black | -0.040 | -0.145, 0.092 |
| Non-Hispanic other or multiple race | 0.001 | -0.018, 0.105 |
| Hispanic | -0.027 | -0.155, 0.157 |
| **Mother is a college graduate** (ref: education less than a college graduate) | *-0.083 | -0.150, -0.016 |
| **Mother never married, widowed, divorced, separated, or deceased** (ref: married) | 0.009 | -0.090, 0.108 |
| **Child's race/ethnicity*mother's education** | | |
| (Ref: non-Hispanic White with college graduate mother) | | |

(Continued)

**Table 3.** (Continued)

| | Up-to-date status: "SERIES BUT NOT FLU" 4:3:1:3:3:1:4 series, not flu | |
| --- | --- | --- |
| | ΔPr. | 95% CI |
| Non-Hispanic Black with college graduate mother | 0.121 | -0.094, 0.336 |
| Non-Hispanic other/multiple race with college graduate mother | 0.058 | -0.122, 0.238 |
| Hispanic with college graduate mother | **0.263 | 0.104, 0.422 |
| **Child's race/ethnicity*mother's marital status** | | |
| (Ref: non-Hispanic White; mother never married, widowed, divorced, separated, or deceased) | | |
| Non-Hispanic Black; mother never married, widowed, divorced, separated, or deceased | 0.022 | -0.157, 0.202 |
| Non-Hispanic other/multiple race; mother never married, widowed, divorced, separated, or deceased | -0.042 | -0.253, 0.169 |
| Hispanic; mother never married, widowed, divorced, separated, or deceased | 0.068 | -0.082, 0.217 |
| **Mother is college graduate*never married/widowed/divorced/separated/deceased** (Ref: mother is college graduate*married) | -0.080 | -0.240, 0.081 |
| **Child's race/ethnicity*mother's education*mother's marital status** | | |
| (Ref: non-Hispanic White; mother is college graduate; never married, widowed, divorced, separated, or deceased) | | |
| Non-Hispanic Black; mother is college graduate; never married, widowed, divorced, separated, or deceased | 0.086 | -0.258, 0.430 |
| Non-Hispanic other/multiple race; mother is college graduate; never married, widowed, divorced, separated, or deceased | 0.076 | -0.357, 0.510 |
| Hispanic; mother is college graduate; never married, widowed, divorced, separated, or deceased | -0.115 | -0.475, 0.244 |
| **Significant covariates** | | |
| Parent ever refused/decided not to have their child vaccinated (ref: never) | **0.099 | 0.042, 0.157 |
| Parent ever delayed or put off having their child vaccinated (ref: never) | **-0.075 | -0.125, -0.026 |

Source: 2011 National Immunization Survey (NIS) data, children represented in the Parental Concerns module with provider-verified vaccination data and eligible for the influenza vaccination up-to-date question who are not missing any covariates. "ΔPr." represents changes in predicted probabilities, weighted to be nationally-representative (e.g., "0.116" means an absolute increase in probability of series but not influenza outcome associated with change in the covariate; this is the same as an 11.6 percentage point absolute increase in chance of series but not influenza outcome associated with change in the covariate). Standard errors used to calculate 95% confidence intervals are adjusted for complex survey design. For brevity, this table only includes the main outcome of interest, main independent variables, and significant covariates. This model controls more many covariates not shown in the table: child sex, child first born status, child WIC recipiency, child insurance status, mother's age group, household language, housing arrangement, provider facility type, child breastfed status, 5 different measures of parental beliefs of perceptions about vaccine and vaccine-preventable diseases, and area of residence. Shaded cells represent significant coefficients indicating vulnerability unique to the "series not influenza" outcome or in a direction different than suggested from the "both" or "neither" outcomes. See S1 Table for the unabridged version with all three outcomes and all covariates.
*p<0.05
**p<0.01 ***p<0.001.

**Table 4. Predicted probabilities of up-to-date vaccine outcomes among intersectional interaction term subgroups, multivariate linear probability model regression, U.S children aged 6–23 months old (N = 7,246), 2011 NIS.**

| | Up-to-date status: "SERIES BUT NOT FLU" 4:3:1:3:3:1:4 series, not flu | |
|---|---|---|
| *Main coefficient subgroups* | Pr. | 95% CI |
| **Child's race/ethnicity** | | |
| Non-Hispanic White only | 0.419 | 0.386, 0.453 |
| Non-Hispanic Black only | 0.434 | 0.358, 0.509 |
| Non-Hispanic other or multiple race | 0.430 | 0.362, 0.498 |
| Hispanic | 0.506 | 0.448, 0.564 |
| **Mother's education** | | |
| Less than a college graduate | 0.448 | 0.414, 0.482 |
| College graduate | 0.429 | 0.378, 0.479 |
| **Mother's marital status** | | |
| Married | 0.427 | 0.395, 0.460 |
| Never married, widowed, divorced, separated, or deceased | 0.424 | 0.374, 0.474 |
| *Two-way interaction term subgroups* | | |
| **Child's race/ethnicity\*mother's education** | | |
| Non-Hispanic White child; non-college graduate mother | 0.452 | 0.407, 0.498 |
| Non-Hispanic White child; college graduate mother | 0.344 | 0.291, 0.396 |
| Non-Hispanic Black child; non-college graduate mother | 0.419 | 0.323, 0.516 |
| Non-Hispanic Black child; college graduate mother | 0.460 | 0.331, 0.589 |
| Non-Hispanic other or multiple race child; non-college graduate mother | 0.439 | 0.335, 0.544 |
| Non-Hispanic other or multiple race child; college graduate mother | 0.413 | 0.284, 0.543 |
| Hispanic child; non-college graduate mother | 0.447 | 0.370, 0.524 |
| Hispanic child; college graduate mother | 0.565 | 0.447, 0.683 |
| **Child's race/ethnicity\*mother's marital status** | | |
| Non-Hispanic White child; married mother | 0.419 | 0.375, 0.464 |
| Non-Hispanic White child; never married, widowed, divorced, separated, or deceased mother | 0.399 | 0.332, 0.466 |
| Non-Hispanic Black child; married mother | 0.423 | 0.320, 0.526 |
| Non-Hispanic Black child; never married, widowed, divorced, separated, or deceased mother | 0.457 | 0.360, 0.553 |

(*Continued*)

**Table 4.** (Continued)

| | Up-to-date status: "SERIES BUT NOT FLU" 4:3:1:3:3:1:4 series, not flu | |
|---|---|---|
| Non-Hispanic other or multiple race child; married mother | 0.441 | 0.346, 0.536 |
| Non-Hispanic other or multiple race child; never married, widowed, divorced, separated, or deceased mother | 0.407 | 0.258, 0.555 |
| Hispanic child; married mother | 0.488 | 0.415, 0.561 |
| Hispanic child; never married, widowed, divorced, separated, or deceased mother | 0.494 | 0.377, 0.611 |
| **Mother's education\*mother's marital status** | | |
| Mother is not a college graduate; married | 0.437 | 0.393, 0.481 |
| Mother is not a college graduate; never married, widowed, divorced, separated, or deceased | 0.463 | 0.411, 0.516 |
| Mother is a college graduate; married | 0.447 | 0.396, 0.498 |
| Mother is a college graduate; never married, widowed, divorced, separated, or deceased | 0.380 | 0.270, 0.490 |
| *Three-way interaction term subgroups* | | |
| **Child's race/ethnicity\*mother's education\*mother's marital status** | | |
| Non-Hisp. White child; mother is not college grad; married | 0.449 | 0.391, 0.508 |
| Non-Hisp. White child; mother is not college grad; never married/widowed/divorced/separated/deceased | 0.458 | 0.382, 0.534 |
| Non-Hisp. White child; mother is college grad; married | 0.366 | 0.319, 0.413 |
| Non-Hisp. White child; mother is college grad; never married/widowed/divorced/separated/deceased | 0.295 | 0.166, 0.424 |
| Non-Hisp. Black child; mother is not college grad; married | 0.409 | 0.276, 0.543 |
| Non-Hisp. Black child; mother is not college grad; never married/widowed/divorced/separated/deceased | 0.441 | 0.351, 0.530 |
| Non-Hisp. Black child; mother is college grad; married | 0.448 | 0.286, 0.609 |
| Non-Hisp. Black child; mother is college grad; never married/widowed/divorced/separated/deceased | 0.485 | 0.275, 0.696 |
| Non-Hisp. other/multiple race child; mother is not college grad; married | 0.450 | 0.308, 0.592 |
| Non-Hisp. other/multiple race child; mother is not college grad; never married/widowed/divorced/separated/deceased | 0.417 | 0.292, 0.542 |
| Non-Hisp. other/multiple race child; mother is college grad; married | 0.425 | 0.333, 0.518 |
| Non-Hisp. other/multiple race child; mother is college grad; never married/widowed/divorced/separated/deceased | 0.388 | 0.040, 0.737 |
| Hispanic child; mother is not college grad; married | 0.423 | 0.325, 0.520 |
| Hispanic child; mother is not college grad; never married/widowed/divorced/separated/deceased | 0.499 | 0.411, 0.588 |
| Hispanic child; mother is college grad; married | 0.603 | 0.489, 0.717 |

*(Continued)*

**Table 4.** (Continued)

| | Up-to-date status: "SERIES BUT NOT FLU" 4:3:1:3:3:1:4 series, not flu | |
|---|---|---|
| Hispanic child; mother is college grad; never married/widowed/divorced/separated/deceased | 0.485 | 0.206, 0.763 |

Source: 2011 National Immunization Survey (NIS) data, children represented in the Parental Concerns module with provider-verified vaccination data and eligible for the "series but not influenza" vaccination up-to-date question who are not missing any covariates from main analysis. Coefficients represent predicted linear probabilities of vaccination up-to-date outcomes among all hierarchical interaction term subgroups from multivariate linear probability regression models (Table 3; i.e., adjusting for all covariates). See S2 Table for the unabridged version with all three up-to-date status outcomes.

among Hispanic children only with married Hispanic mothers. Hispanic mothers not in the married group appear to have the same education interaction as non-Hispanic White and non-Hispanic Other/multiple race children. The direction of the interaction term coefficient compared to its interaction term coefficient in the "both" or "neither" columns of Table 3 suggests this is unique to "series but not influenza" vulnerability.

## Discussion

A concerning main finding of this study is that nearly half of very young US children have "hidden vulnerability to influenza." These children are UTD on a large series of vaccine recommendations (a 19-shot, 7-vaccine series)–and would otherwise seem like neither a population vulnerable to vaccine-preventable diseases nor suggest their parents would have tendencies to refuse vaccination–but yet are not UTD on influenza vaccinations. A recent study of complete influenza vaccine uptake among very young NIS children found nearly identical uptake [24] as reported here, though differences in respondents' intent to receive other vaccines and the role that parental attitudes toward vaccination and vaccine-preventable diseases were not studied. We were able to examine this finding including comparisons to both

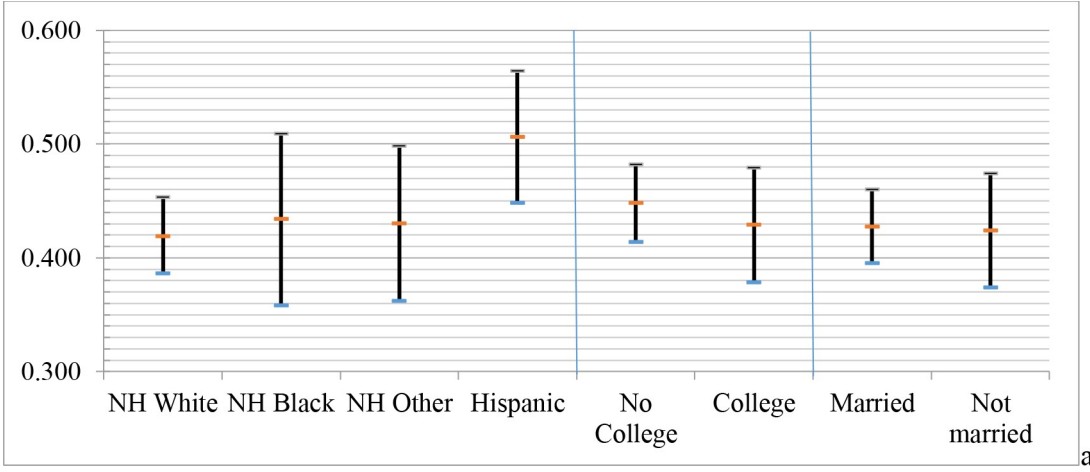

**Fig 1. Model-predicted probability (with 95% confidence intervals) of "series but not flu" outcome among main coefficient subgroups from Table 4.**

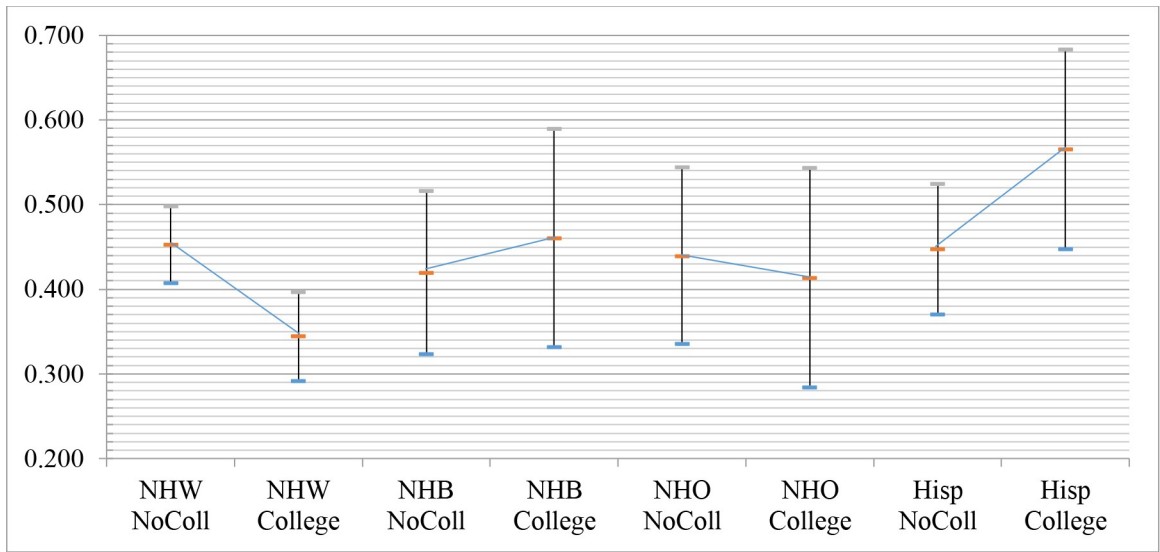

**Fig 2. Model-predicted probability (with 95% confidence intervals) of "series but not flu" outcome among two-way interaction term subgroups: Child's race/ethnicity\* mother's education from Table 4.** Note the upward slanting slopes of "series but not flu" probability among non-Hispanic Black and Hispanic children when their mothers had a college education.

uptake of other vaccines and adjusting for parental attitudes toward vaccination and vaccine-preventable diseases.

Parental history of vaccine refusal was unsurprisingly associated with lower UTD status of all vaccines studied (the 4:3:1:3:3:1:4 and complete influenza vaccine status). What is particularly interesting, however, is that a unique determinant of hidden vulnerability to influenza was parental history of never delaying vaccination. While vaccine hesitancy has risen recently, [65] child influenza vaccination rates have been lower than other vaccines for quite some time and our finding was independent of general vaccine hesitancy. This finding likely represents longstanding hesitancy specific to the influenza vaccine.

Perhaps many parents with children UTD on most vaccines, who thus appear to support the concept of vaccination, are uniquely hesitant or skeptical about the influenza vaccine. This supports the theory that vaccine hesitancy is highly context-dependent and functions differently comparing influenza to other vaccines. Vaccine hesitancy is complex; it is heavily grounded in myths about vaccines and their respective diseases, as well as interwoven with broader contexts such as socioeconomic circumstances, social norms, health beliefs, the media, and institutional trust. [65–69]

The second unique predictor of hidden vulnerability to influenza was maternal college education attainment (but only for non-Hispanic Black children, and Hispanic children with married mothers, suggesting that intersectionality is important to identifying hidden vulnerability to influenza). In other words, maternal college degree attainment was associated with higher uptake of all vaccines studied *except* among non-Hispanic Black and Hispanic children, for whom it was instead associated with "hidden vulnerability" to influenza.

Higher parental education is generally associated higher vaccine uptake in US children, [41–43] though the returns of higher education may differ by race/ethnicity, particularly with regards to health behavior. [70] Intersectionality is a fundamental concept not just as it pertains to social disadvantage but also as it pertains to health, [32–35] yet has unfortunately been largely neglected in the health literature. [31] Public health and health policy researchers have placed increasing recognition on the notion that health equity can only occur by incorporating

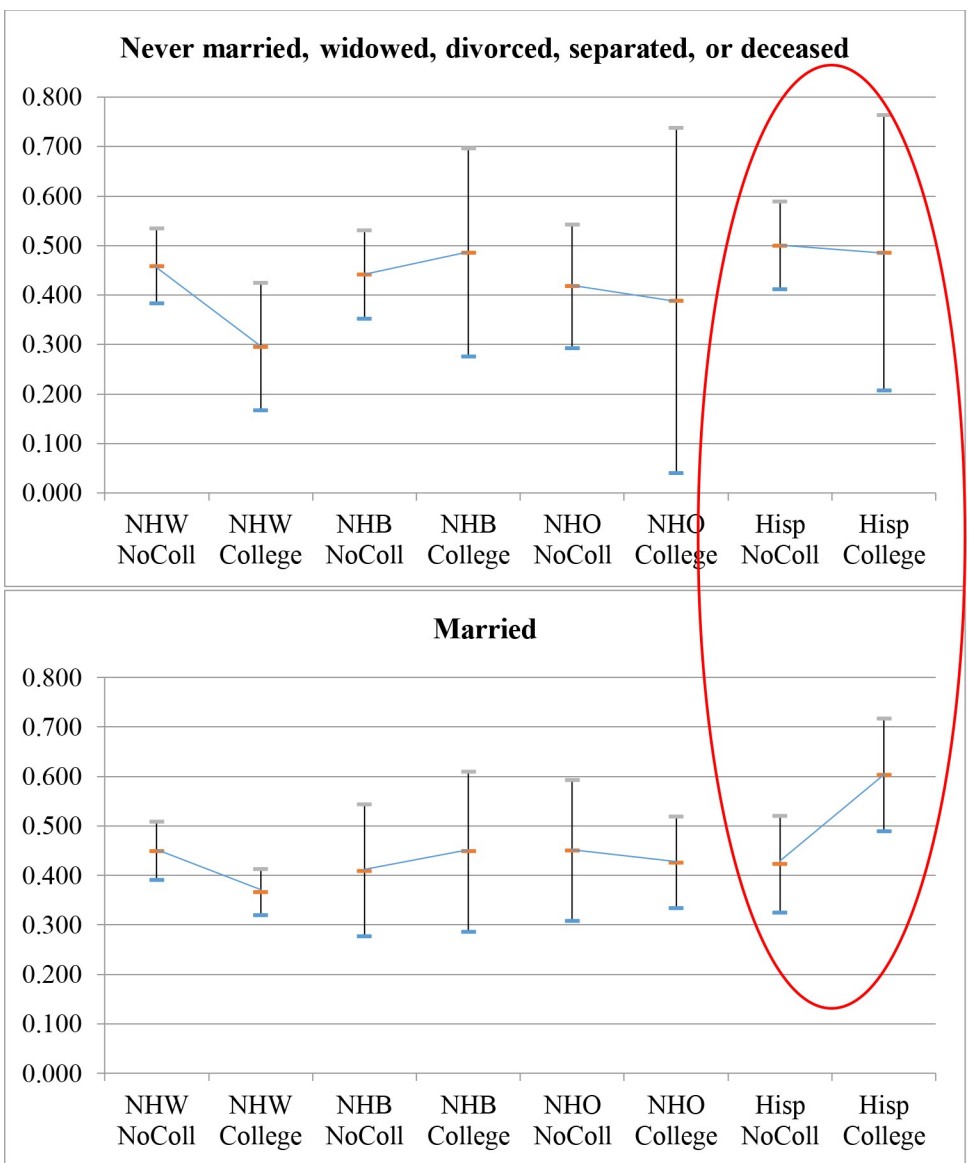

**Fig 3. Model-predicted probability (with 95% confidence intervals) of "series but not flu" outcome among three-way interaction term subgroups: Child's race/ethnicity*mother's education, stratified by mother's marital status from Table 4.** Note that all trend lines in the top graph parallel trend lines in the bottom graph except those circled in red.

health into upstream decision-making, such as social and economic policy (e.g., the "Health in All Policies" approach). [71] This study reinforces these points and criticisms coming from both sociologists and public health professionals, as the intersectionality of maternal education and child's race/ethnicity revealed disparities not observed when examining them individually.

These findings should be interpreted within this study's limitations. First, the influenza vaccine UTD variable does not capture vaccinations after December 31st or through the date of the interview (first dose), or after January 31st (second dose), [38] though influenza vaccine distribution is usually complete before these dates, [72] meaning that this limitation is minor. Further, the provider-verified nature of the NIS complete vaccination outcome improves on

the typical annual self-reported measure of influenza vaccination, which is subject to recall bias and only covers one influenza season. Second, this study excludes children without provider-verified data, who may lack this type of data because they lack a usual source of care, which has been linked to lower preventive care use in adults. [73] However, because those excluded may use less preventive services, the implication is that our findings contain less vulnerable individuals and are likely thus conservative. Third, accounting for successive non-response first from households, then providers, and then the PC module, more than half of target children are lost due to NIS non-response issues, introducing concerns of non-response bias. This is a limitation of the data source itself that warrants investigation and needs to be addressed in future surveys. Nonetheless, the NIS still provides the only opportunity to examine nationally-representative, provider-verified uptake of multiple vaccines in young children that includes key constructs for vaccine-related parental perceptions. Fourth, the parental concerns variables refer to vaccination generally and not to any one specific vaccine, which could explain some of the non-findings (such as parent perception of physician recommendation for vaccination not being associated with our outcomes, contradicting other studies [27,41,45–50,52–56]). Fifth, this analysis is cross-sectional and thus cannot make causative claims; all findings are associative. That said, the main identifying strategies were to use only provider-verified vaccine outcomes and to include in one model a myriad of conceptually- and empirically-grounded covariates more comprehensive than in other literature, most notably the aforementioned constructs for vaccine-related parental perceptions which have seldom been utilized due to their limited availability and the restricted access required to obtain them. Though we cannot rule out the possibility of bi-directionality in our findings, we believe this to be less likely as the determinants studied here are thought to temporally precede the decision to use a health service. [39] For example, predisposing (child's race/ethnicity) and enabling factors (mother's education) precede personal health services use factors at the behavior level (history of vaccine refusal or delay), all of which precede health services utilization (vaccine uptake).

This study provides important findings and data regarding "hidden vulnerability to influenza"–a phenomenon whereby nearly half (44%) of very young US children are up-to-date on a large series of routinely-recommended vaccines yet are not UTD against influenza by their second birthday–despite high morbidity of influenza in this age group. Independent of an expansive set of confounders, the most important factor predicting vaccine vulnerability is history of vaccine refusal, though there was also an independent, unique association of hidden vulnerability to influenza with having never delayed vaccination.

Healthcare clinicians need to have conversations surrounding vaccine hesitancy even with parents of children who appear to be broadly up-to-date on their vaccines and thus appear to generally support the concept of vaccination. These parents are unlikely to give any indication of their skepticism of influenza vaccines yet this study finds that they may opt to not have their child vaccinated against influenza. Pediatricians and other healthcare clinicians who see children should consider adding questions to their history and physical protocols pertaining to parental history of refusing or delaying vaccination, as well as pertaining to vaccine hesitancy both broadly and specifically to influenza regardless of the child's general vaccine history.

Further, this study suggests that parental college education and marriage may not translate into improved influenza vaccine uptake for children of historically-disadvantaged race/ethnicity despite that it does for non-Hispanic White children. Policymakers and researchers from public health, sociology, and other sectors need to collaborate to examine both how preventive health services use functions in the context of interacting social disadvantage, and how upstream social and economic policies lead to equitable health.

## Supporting information

**S1 Table.**
(DOCX)

**S2 Table.**
(DOCX)

## Acknowledgments

First, we acknowledge Rhonda BeLue, Steven A. Haas, and Marianne H. Hillemeier from Pennsylvania State University for helping to progress earlier versions of this work. The authors would also like to thank Sarah E. Patterson for bringing to our attention the "Opening Influenza Research Project" opportunity, of which this article is a part; the opportunity helped us bring this article out of our "file drawers" and to see it to publication.

Second, we acknowledge Patricia Barnes of the National Center for Health Statistics (NCHS), and Emily Greenman, and Mark Roberts of the Penn State Federal Statistical Research Data Center (RDC) for helping to review proposals and access restricted data from a related project used to inform this work.

Third, data collection for National Health Interview Survey and the National Immunization Survey, analyzed in this work, was approved by the NCHS Research Ethics Review Board (ERB). Analysis of de-identified data from the survey is exempt from the federal regulations for the protection of human research participants. Analysis of restricted data through the NCHS Research Data Center is also approved by the NCHS ERB. The findings and conclusions in this work are those of the authors and do not necessarily represent the views of the Research Data Center, the National Center for Health Statistics, or the Centers for Disease Control and Prevention.

Fourth, this work was completed while William K. Bleser was at the Department of Health Policy and Administration, Pennsylvania State University, University Park, Pennsylvania.

## Author Contributions

**Conceptualization:** William K. Bleser, Patricia Y. Miranda.

**Data curation:** William K. Bleser.

**Formal analysis:** William K. Bleser.

**Funding acquisition:** William K. Bleser, Patricia Y. Miranda.

**Methodology:** William K. Bleser, Daniel A. Salmon, Patricia Y. Miranda.

**Project administration:** William K. Bleser, Patricia Y. Miranda.

**Supervision:** Daniel A. Salmon, Patricia Y. Miranda.

**Visualization:** William K. Bleser.

**Writing – original draft:** William K. Bleser.

**Writing – review & editing:** William K. Bleser, Daniel A. Salmon, Patricia Y. Miranda.

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
