## [Decision Letter · Decision Letter 0]

11 Mar 2020

PONE-D-19-33172

A hidden vulnerable population: young children up-to-date on vaccine series recommendations except influenza vaccines

PLOS ONE

Dear Dr Bleser,

Thank you for submitting your manuscript to PLOS ONE. After careful consideration, we feel that it has merit but does not fully meet PLOS ONE’s publication criteria as it currently stands. Therefore, we invite you to submit a revised version of the manuscript that addresses the points raised during the review process.

While I think this is an important topic that warrants investigation, there were several issues with both the study design, and the manuscript itself that are significant enough that they seriously undermine the contributions of the study. The manuscript has a number of weaknesses which need to be considered.

The PLOS ONE publish research on the basis of scientific validity and rigorous methodology. Together with both reviewers I have a number of reservations about this paper regarding both above mentioned issues. They are outlined below.

According to the journal criteria  an analytical, cross-sectional study, like this one must has been conducted rigorously. However, the methods used are not clearly presented.  The work contains some methodological mistakes which  diminish its quality. I suggest to reconsider the methods used (see the list of reviewers’ comments below).

We would appreciate receiving your revised manuscript by March 15th. To enhance the reproducibility of your results, we recommend that if applicable you deposit your laboratory protocols in protocols.io, where a protocol can be assigned its own identifier (DOI) such that it can be cited independently in the future. For instructions see: http://journals.plos.org/plosone/s/submission-guidelines#loc-laboratory-protocols

We look forward to receiving your revised manuscript.

Kind regards,

Prof. Maria Gańczak

Academic Editor

PLOS ONE

Journal Requirements:

"We acknowledge Pennsylvania State University's Department of Health Policy and Administration, Demography program, and Population Research Institute for supporting this research. The Population Research Institute is supported by an infrastructure grant from NIH (2R24HD041025-11).

This publication was also supported, in part, by Grant UL1 TR000127 and KL2 TR000126 from the National Center for Advancing Translational Sciences (NCATS)."

"William K. Bleser discloses consulting fees from Merck unrelated to this research. Daniel A. Salmon discloses consulting fees and research grants from Merck, Pfizer, and Walgreens unrelated to this research. No other financial disclosures were reported."

We note that one or more of the authors are employed by a commercial company: Merck, Pfizer, and Walgreens

4. We noted in your submission details that a portion of your manuscript may have been presented or published elsewhere. ["An earlier version of this manuscript was published as a part of a dissertation. "] Please clarify whether this [conference proceeding or publication] was peer-reviewed and formally published. If this work was previously peer-reviewed and published, in the cover letter please provide the reason that this work does not constitute dual publication and should be included in the current manuscript.

5. Your ethics statement must appear in the Methods section of your manuscript. If your ethics statement is written in any section besides the Methods, please move it to the Methods section and delete it from any other section. Please also ensure that your ethics statement is included in your manuscript, as the ethics section of your online submission will not be published alongside your manuscript.

Reviewers' comments:

Reviewer's Responses to Questions

**Comments to the Author**

1. Is the manuscript technically sound, and do the data support the conclusions?

Reviewer #1: Partly

Reviewer #2: Yes

2. Has the statistical analysis been performed appropriately and rigorously? 

Reviewer #1: Yes

Reviewer #2: I Don't Know

3. Have the authors made all data underlying the findings in their manuscript fully available?

Reviewer #1: Yes

Reviewer #2: Yes

4. Is the manuscript presented in an intelligible fashion and written in standard English?

Reviewer #1: Yes

Reviewer #2: Yes

5. Review Comments to the Author

Reviewer #1: This manuscript presents an analysis of nationally representative survey data on influenza vaccine uptake in young children. The authors investigate cross-sectional associations with a number of predictors of vaccination, including social and demographic variables and past vaccine utilization. Importantly, the analysis also seeks to identify intersectionality, which is a novel and important approach that represents a significant contribution to the literature on vaccine hesitancy. Overall, the analysis is rigorous, the conclusions are supported by the data, and the manuscript is well-written and clear. I agree with the authors' conclusion that it is concerning that there may influenza vaccine specific hesitancy, even among parents who are regularly choosing vaccination for their children to prevent other diseases. I have some questions and comments, which I believe would strengthen the manuscript. These comments are separated by manuscript section below.

Introduction

It is important that the authors are clear about their assertions about risk of influenza and benefits of influenza vaccine. Specifically, children < 5 years old are typically considered to be at lower risk of influenza infection than school aged children (most likely due to to contact patterns), unless they attend child care outside of the home. The references cited here are reporting risk of complications due to influenza infection (e.g. pneumonia) which children <5 ARE considered to be at especially high risk for developing. I suggest a minor change to the language to indicate that these children are at higher risk of complications, rather than saying they are simply at high risk of influenza. Similarly, influenza vaccination is the most effective preventive measure for prevention of infection,

Methods:

The scale for parent belief variables should be stated, otherwise the mean and SD values reported in table 1 are difficult to interpret for someone who is not familiar with these data.

The primary outcome of interest is potentially problematic due to the annual nature of influenza vaccines. Unless I am misunderstanding this outcome, a 2 year old child who received two doses of influenza vaccine in the year of the survey (2011) but was unvaccinated in the previous year (2010) would be considered to have "hidden vulnerability" to influenza. This is true, even though, by ACIP recommendations that child would be considered fully vaccinated against influenza for the 2011 season. In addition, that child may actually have higher antibody titres to vaccine strains and be expected to have higher vaccine effectiveness (due to negative interference) for the 2011 season. At a minimum, I recommend a sensitivity analysis where children are classified as "fully vaccinated", "partially vaccinated" or "unvaccinated" against influenza using ACIP guidelines to determine this categorization.

Was there any correction for multiple testing? Why or why not?

Results:

The tables are difficult to read and interpret due to the large number of variables and interactions tested. I would recommend that the authors present interesting findings and relevant supporting information in the main tables and move everything else to supplemental tables. Another possibility is to present only the "series but not flu" outcome in the main tables and move other outcomes to a supplementary table.

I find the predicted probabilities much easier to interpret than the the beta coefficients, and suggest focusing on those results. The figures are particuarly helpful.

Discussion:

The strengths and limitations are adequately discussed.

Reviewer #2: This is a generally well written manuscript describing what appears to be a very well designed and conducted study (although I am unable to comment definitively on the appropriateness/rigour of the statistical analyses as this is not my particular area of expertise) exploring an important area where previously published research is limited, and with findings that should be of broad interest. A few suggestions to enhance the paper are provided.

Introduction

The sentence “Further, “complete uptake” – receiving the appropriate number of influenza vaccinations for the child’s age – is much lower, peaking at 45% in children 6-23 months old” is confusing. Children aged <9 years require 2 doses in their first season of vaccination so the number of doses of influenza vaccination recommended in any particular season depends on both the age and number of doses previously received. The paper cited looked at children aged 6-23 months from 2002 to 2012 and found that complete vaccination in this age group peaked at 45% in 2011/12, which is a quite different conclusion from that noted above. Complete vaccination for children aged 2-<5 years, who will often only need one dose, will be higher. Suggest reword this sentence.

Methods

It is not clear how ‘complete influenza vaccination’ has been assessed. This is described as “whether the child received the full number of influenza vaccines given the number of influenza seasons they have experienced by their second birthday”. However as noted above this depends on both age and number of doses previously received. In this age group, if assessing ‘complete influenza vaccination’ for the most recent influenza season this would require two doses to have been received unless two doses had been received in the previous season, in which case would only require one dose. Suggest reword to make clearer how assessed.

Discussion

2nd para states that the unique determinant of hidden vulnerability to flu identified (parental history of never delaying vaccination) “may reflect the rise in vaccine hesitancy”. However, influenza vaccination rates in children have always been lower than other vaccines, likely reflecting longstanding hesitancy specific to this vaccine. Suggest delete this wording – the remainder of the relevant sentence stands alone without this somewhat dubious assertion.

There is much mention of parental attitudes to vaccination however no mention of potential impact of provider attitudes to influenza vaccination. I am unaware what research has been conducted in this area in the US, but there is certainly evidence from other countries that ‘hesitancy’ amongst providers regarding influenza vaccination is higher than for other vaccines, particularly related to the comparatively low effectiveness of influenza vaccine, which may impact on recommendation/ strength of recommendation. This is important given the extensive evidence that strong provider recommendation is a key determinant of vaccine uptake. While parent perception of strength of physician’s vaccine recommendation was included as a variable in this study, with no significant associations detected, I assume that this was framed in the survey in general terms rather than specific to particular vaccines such as influenza. Suggest add brief discussion of this issue to the limitations section.

General comment – suggest a thorough proof read as there are a moderate number of typographical and grammatical errors.

6. PLOS authors have the option to publish the peer review history of their article (what does this mean?). If published, this will include your full peer review and any attached files.

Reviewer #1: No

Reviewer #2: Yes: Frank Beard

---

## [Author Response · Author response to Decision Letter 0]

17 Apr 2020

See submitted "Response to Reviewers" document.

---

## [Decision Letter · Decision Letter 1]

28 May 2020

A hidden vulnerable population: young children up-to-date on vaccine series recommendations except influenza vaccines

PONE-D-19-33172R1

Dear Dr. Bleser,

We are pleased to inform you that your manuscript has been judged scientifically suitable for publication and will be formally accepted for publication once it complies with all outstanding technical requirements.

With kind regards,

Prof. Maria Gańczak

Academic Editor

PLOS ONE

Additional Editor Comments (optional):

Reviewers' comments:

Reviewer's Responses to Questions

**Comments to the Author**

1. If the authors have adequately addressed your comments raised in a previous round of review and you feel that this manuscript is now acceptable for publication, you may indicate that here to bypass the “Comments to the Author” section, enter your conflict of interest statement in the “Confidential to Editor” section, and submit your "Accept" recommendation.

Reviewer #1: All comments have been addressed

Reviewer #2: All comments have been addressed

2. Is the manuscript technically sound, and do the data support the conclusions?

Reviewer #1: Yes

Reviewer #2: (No Response)

3. Has the statistical analysis been performed appropriately and rigorously? 

Reviewer #1: Yes

Reviewer #2: (No Response)

4. Have the authors made all data underlying the findings in their manuscript fully available?

Reviewer #1: Yes

Reviewer #2: (No Response)

5. Is the manuscript presented in an intelligible fashion and written in standard English?

Reviewer #1: Yes

Reviewer #2: (No Response)

6. Review Comments to the Author

Reviewer #1: The authors responded to each comment in a satisfactory manner. I have no additional comments, and believe the manuscript is not suitable for publication.

Reviewer #2: (No Response)

7. PLOS authors have the option to publish the peer review history of their article (what does this mean?). If published, this will include your full peer review and any attached files.

Reviewer #1: No

Reviewer #2: Yes: Frank Beard

---

## [Editor Report · Acceptance letter]

5 Jun 2020

PONE-D-19-33172R1 

A hidden vulnerable population: young children up-to-date on vaccine series recommendations except influenza vaccines 

Dear Dr. Bleser:

I'm pleased to inform you that your manuscript has been deemed suitable for publication in PLOS ONE. Congratulations! Your manuscript is now with our production department. 

Kind regards, 

on behalf of

Prof. Maria Gańczak 

Academic Editor

PLOS ONE